# Freshwater Supply to Metropolitan Shanghai: Issues of Quality from Source to Consumers

**Maotian Li [1], Jing Chen [1,\*], Brian Finlayson [2], Zhongyuan Chen [1], Michael Webber [2] , Jon Barnett [2] and Mark Wang [2]**

[1] State Key Laboratory of Estuarine and Coastal Research, East China Normal University, Shanghai 200062, China; mtli@sklec.ecnu.edu.cn (M.L.); z.chen@ecnu.edu.cn (Z.C.)

[2] School of Geography, University of Melbourne, Carlton 3010, Australia; brianlf@unimelb.edu.au (B.F.); mjwebber@unimelb.edu.au (M.W.); jbarn@unimelb.edu.au (J.B.); myw@unimelb.edu.au (M.W.)

\* Correspondence: jchen@geo.ecnu.edu.cn

**Abstract:** Shanghai is experiencing drinking water supply problems that are caused by heavy pollution of its raw water supply, deficiencies in its treatment processes, and water quality deterioration in the distribution system. However, little attention has been paid these problems of water quality in raw water, water treatment, and household drinking water. Based on water quality data from 1979 to 2016, we show that microbes (TBC), eutrophication (TP, TN, and $NH_3$–N), heavy metals (Fe, Mn, and Hg), and organic contamination (chemical oxygen demand (COD), detergent (Linear Alklybenzene Sulfonate, LAS), and volatile phenols (VP)) pollute the raw water sources of the Huangpu River and the Changjiang (Yangtze River) estuary. The average concentrations of these contaminants in the Huangpu River are almost double that of the Changjiang estuary, forcing a rapid shift to the Changjiang estuary for raw water. In spite of filtering and treatment, TN, $NH_3$–N, Fe, COD, and chlorine maxima of the treated water and drinking water still exceed the Chinese National Standard. We determine that the relevant threats from the water source to household water in Shanghai are: (1) eutrophication arising from highly concentrated TN, TP, COD, and algal density in the raw water; (2) increasing salinity in the river estuary, especially at the Qingcaosha Reservoir (currently the major freshwater source for Shanghai); (3) more than 50% of organic constituents and by-products remain in treated water; and, (4) bacteria and turbidity increase in the course of water delivery to users. The analysis presents a holistic assessment of the water quality threats to metropolitan Shanghai in relation to the city's rapid development.

**Keywords:** Shanghai; water quality; eutrophication; conventional water treatment; secondary water pollution

## 1. Introduction

There is no substitute for freshwater and it plays an essential role in sustaining societies and supporting and enriching the culture [1,2]. Pollution from domestic, industrial, and agricultural activities has led to the deterioration of water quality and exacerbated the problems of the availability of good quality water, especially in downstream coastal urban areas and in water-scarce areas [3,4]. Different countries or regions, depending on the purpose for which it are used, such as domestic consumption, industrial use, and environmental systems, usually categorize water contamination [5]. In China, the quality of surface freshwater is assessed based on 29 contaminants and is then classified in five grades (I–V): grades I–III are suitable to be treated for drinking water, IV can be used by industry, and V is only suitable for agricultural or environmental use [6]. Water that fails to reach even the quality required for category V is classed as not being suitable for any functional use.

However, in 2012, 32.3% of all China's freshwater sources (river and lake) were categorized as worse than grade III and could not be used as the source for human drinking water [7]. In addition, more than 70% of the river reaches in urban areas had water quality of grade IV or worse [7]. As a consequence, 300 cities (45.3% of all China's cities) had severe water shortages due to pollution, and they had to build large and complex systems to purify polluted water [8], consisting of three subsystems: raw water plants, water treatment plants, and water delivery systems [9].

Shanghai residents recognize that the piped water supply is substandard and they do not drink it directly from the tap. In a survey that was based on a stratified random sampling procedure conducted in 2013 by the University of Melbourne and East China Normal University, only three of the 4967 respondents said that they would drink the water directly from the public supply system. Residents clearly do not trust the water treatment system and boil the water before use or use commercially available bottled water [10,11].

While Shanghai has access to an abundant supply of water, its use is limited, as much of the local surface water is grade V or worse [12]. Shanghai has a large water treatment and supply system to meet the freshwater demands of its 24 million inhabitants. More importantly, Shanghai is one of the most developed cities in China and, while it has abundant primary water quality data recorded at raw water plants and treatment plants, timely access to this data is restricted. In this paper, on the basis of the data that are available to us for the period from 1979 to 2016, we attempt a holistic assessment of water quality and supply in metropolitan Shanghai. We identify the key issues and risks to water quality from raw water sources to household drinking water. From this, we discuss the potential for improving drinking water quality and provide a prototype reference for better management of raw water sources and supply for the megacity of Shanghai (Figure 1B).

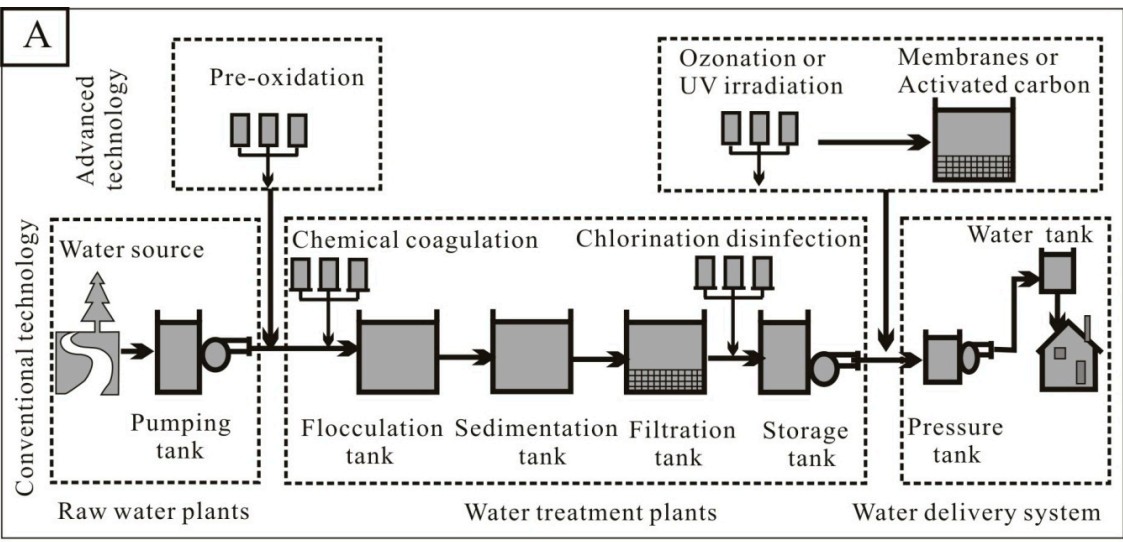

**Figure 1.** *Cont.*

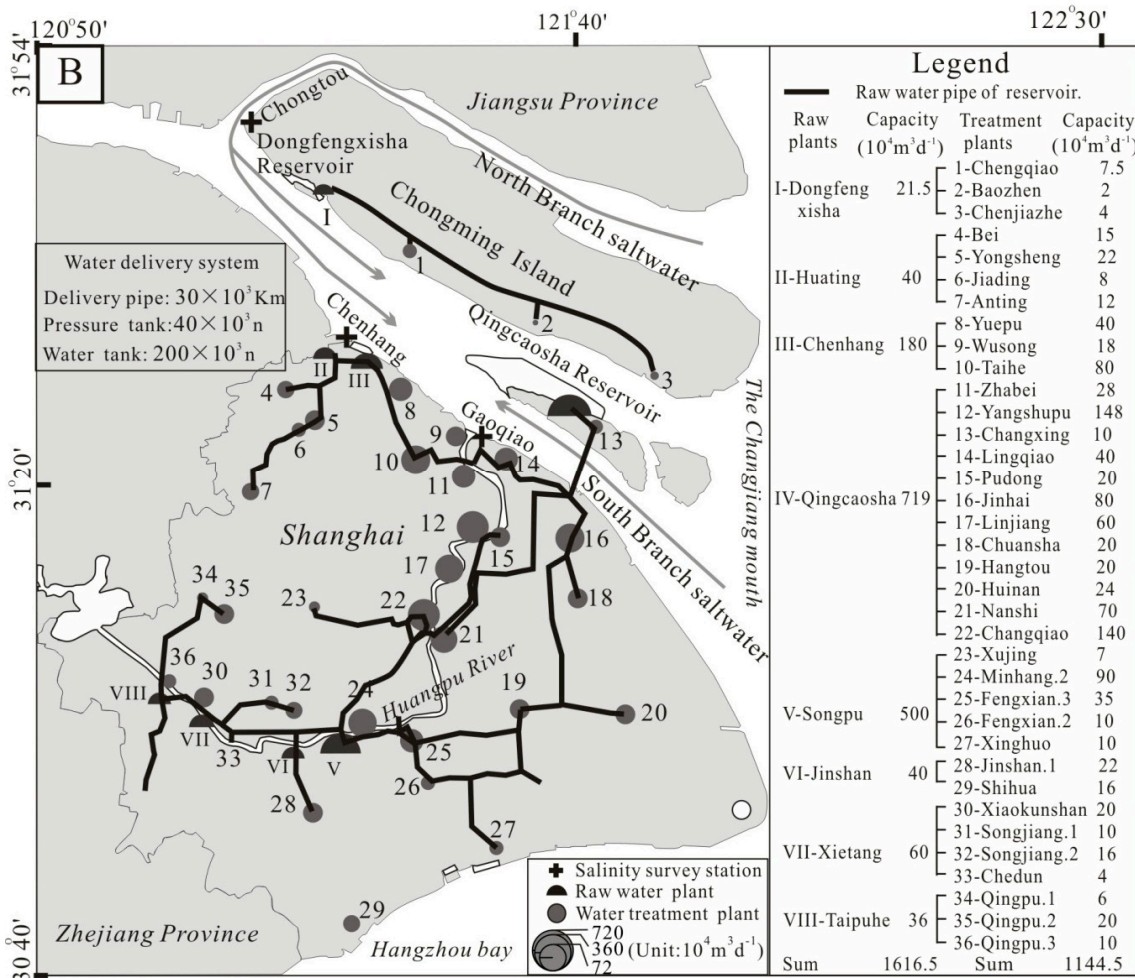

**Figure 1.** (**A**) Flow chart of the water supply system in Shanghai, (**B**) the distribution of raw water plants, water treatment plants and water delivery pipes in Shanghai in 2013, and the sites where water quality and threat factor data were collected.

## 2. Shanghai Water Supply System

Shanghai, a city of 24 million, the largest in China, covers 6340.5 square kilometers and it is situated on a peninsula on the east coast of China between Hangzhou Bay and the Changjiang Estuary (Figure 1B). The water supply system of Shanghai consists of three sub-systems: raw water plants, water treatment plants, and water delivery system installed at point of use (Figure 1A). In 2013, the raw water subsystem contained eight municipal raw water plants (I–VIII, Figure 1B) with a total daily capacity of $16.17 \times 10^6$ m$^3$; the water treatment subsystem included 36 public waterworks (1–36, Figure 1B) with a daily capacity of $11.45 \times 10^6$ m$^3$; and, the drinking water subsystem consisted of a $30 \times 10^3$ km pipe network, $40 \times 10^3$ pressure tanks, and $200 \times 10^3$ building water tanks (Figure 1B). There are eight water intake pumps: four (Songpu, Jinshan, Xietang, and Taipuhe) with a total capacity of $6.36 \times 10^6$ m$^3 \cdot$d$^{-1}$ (39.3% of the total withdrawal capacity) are located on the upper Huangpu River, and four (Qingcaosha, Chenhang, Huating, and Dongfengxisha) with a total capacity of $9.81 \times 10^6$ m$^3 \cdot$d$^{-1}$ (60.7% of the total withdrawal capacity) are located on the Changjiang Estuary (Figure 1B).

The water treatment plants are located in four zones on the basis of different raw water sources. The Dongfengxisha zone, located on Chongming Island, consisting of three urban public water plants (1 to 3, Figure 1B), with a total capacity of only $0.14 \times 10^6$ m$^3 \cdot$d$^{-1}$ (1.2% of the total supply capacity of Shanghai) sources its raw water from Dongfengxisha Reservoir (Figure 1B); the Chenhang zone consisting of seven urban public water plants (4 to 10, Figure 1B) with a total capacity of

$1.95 \times 10^6$ m$^3 \cdot$d$^{-1}$ (13.8% of the total supply capacity of Shanghai) sources its freshwater from the Chenhang and Huating reservoirs sited in the upper Changjiang estuary (Figure 1B); the Qingcaosha zone consisting of 12 urban public water plants (11 to 22, Figure 1B) with a total capacity of $6.60 \times 10^6$ m$^3 \cdot$d$^{-1}$ (57.6% of the total supply capacity) sources its freshwater from the Qingcaosha Reservoir on the Changjiang Estuary (Figure 1B); the Huangpu River zone that consists of 14 urban public water plants (23 to 36, Figure 1B) with a total capacity of $2.76 \times 10^6$ m$^3 \cdot$d$^{-1}$ (19.5% of the total supply capacity) sources raw water from Songpu, Jinshan, Xietang, and Taipuhe plants, all sited on the Huangpu River (Figure 1B).

## 3. Data Sources and Methods

The location and capacity of raw water plants, water treatment plants and the length of water supply pipes, numbers of pressure tanks, and building tanks were all extracted from local government documents and the online database of the Office of Shanghai Local Chronicles [12,13].

The Chinese national water quality standards [6,14] cover 29 contaminants of raw water and 106 contaminants of treated and drinking water. 12 priority contaminants of raw water, and eight priority contaminants of treated and drinking water were selected on the basis of their severity rating and the health risks, as given by the World Health Organisation [15]. The 12 priority contaminants of raw water consist of a microbial measure (total bacteria count; TBC) and 11 chemical components: total phosphorus (TP), total nitrogen (TN), ammonia nitrogen (NH$_3$–N), iron (Fe), manganese (Mn), chemical oxygen demand (COD), dissolve oxygen (DO), mineral oil (MO), detergent (LAS), volatile phenols (VP), and mercury (Hg). In addition to TBC, NH$_3$–N, Fe, Mn, COD, the priority contaminants of treated and drinking water further include two acceptability aspects (chromaticity and turbidity) and one by-product (chlorine) (Figure 2). The details of the data sources are given in Table 1. All priority water contaminants were presented in Figure 2.

**Table 1.** Data sources for water quality contaminants and threat factors for raw water, treated water and drinking water in Shanghai. Where the sampling frequency is given as 'annual' this is the mean of samples collected over the year.

| Database Name | Water Type | Data | Water Source | Station | Time | Sampling Frequency | Data Source |
|---|---|---|---|---|---|---|---|
| Water quality indicators | Raw water | 12 indicators (Figure 2) | YRE | QR | IV | 2010–2011 | annual | [16,17] |
| | | | | CR | III | 1992–1996 | annual | [18] |
| | | | | | III | 2005–2009 | annual | [19] |
| | | | | UHR | V | 1991–1996 | annual | [20] |
| | | | | | V | 1998–2003 | annual | [21] |
| | | | | | V | 2005–2009 | annual | [19] |
| | Treatment water & drinking water | 8 indicators (Figure 2) | YRE | QR | 15, 16 | 2010–2011 | annual | [16] |
| | | | | | 15, 16 | 2012 | annual | [22] |
| | | | | | 15, 16, 17 | 2013–2016 | annual | [23] |
| | | | | CR | 8, 9, 10 | 2003–2004 | annual | [24] |
| | | | | | 8, 9, 10 | 2007 | annual | [25] |
| | | | | | 8, 9, 10 | 2012 | annual | [22] |
| | | | | | 8, 9, 10 | 2013–2016 | annual | [23] |
| | | | | UHR | 23, 23, 24 | 1991–1996 | annual | [20] |
| | | | | | 17, 22 | 1996–1998 | annual | [26] |
| | | | | | 23, 23, 24 | 2003–2004 | annual | [24] |
| | | | | | 23, 23, 24 | 2005 | annual | [27] |
| | | | | | 23, 23, 24 | 2007 | annual | [25] |
| | | | | | 23, 23, 24 | 2012 | annual | [22] |
| | | | | | 31, 32, 33 | 2013–2016 | annual | [23] |

**Table 1.** *Cont.*

| Database Name | Water Type | Data | | Water Source | Station | Time | Sampling Frequency | Data Source |
|---|---|---|---|---|---|---|---|---|
| Treatment factors | Raw water | Eutrophication | TN, TP, COD | YRE | III | 1984–2012 | annual | [18,28,29] |
| | | | | UHR | V | 1986–2016 | annual | [30] |
| | | | Algae | YRE | III | 2002–2003 | daily | [19] |
| | | | | | IV | 2010 | daily | [31] |
| | | | | | IV | 2010–2011 | daily | [32] |
| | | | | UHR | V | 2004–2005 | | [19] |
| | | Salinity | | YRE | Chongtou | 1979–2011 | daily | [33–40] |
| | | | | | Chenhang | 1979–2011 | daily | |
| | | | | | Gaoqiao | 1979–2011 | daily | |
| | Treatment water | Remove rate of 14 indicators | | CR | 8 | 1992–1996 | annual | [18] |
| | | | | UHR | 24 | 1991–1996 | annual | [20] |
| | | | | | 24 | 2005 | annual | [41] |
| | Drinking water | Deterioration of 4 indicators | | UHR | 24 | 2005 | daily | [41] |

Notes: YRE—the Yangtze River Estuary, QR—the Qingcaosha Reservoir, CR—the Chenhang Reservoir, UHR—the Upper Huangpu River; Location in Figure 1.

We selected four eutrophic factors, TN, TP, COD, and algal density, for threat factors in raw water to assess the level of threats of raw water, because they serve as essential constituents that drive up other microbial and chemical aspects [42–44]. Salinity data from three hydrological stations in the Changjiang estuary (Gaoqiao, Chenhang and Chongtou, Figure 1) is also presented, because reservoirs in the estuary suffer events of saltwater intrusion when salinity exceeds 0.45 practical salinity units (psu) and it is not suitable for treatment [45]. Table 1 describes the data sources for eutrophication and salinity of raw water.

Removal rates (%) of fourteen contaminants (chromaticity and turbidity, TBC, TP, TN, $NH_3$–N, Fe, Mn, COD, LAS, VP, chlorine, chloroform, and carbon tetrachloride ($CCl_4$)) were selected from local water plants to assess the quality of treated water (Table 1). For assessing the quality of water delivered for drinking and household use, 4 contaminants (chlorine, TBC, COD and turbidity) were collected from two papers [20,41] (Table 1).

By comparing all of the key contaminants of raw water, treated water, and drinking water based on the national standards of surface water [6] and drinking water [14], we elaborate on the risks and vulnerability of the three water supply subsystems.

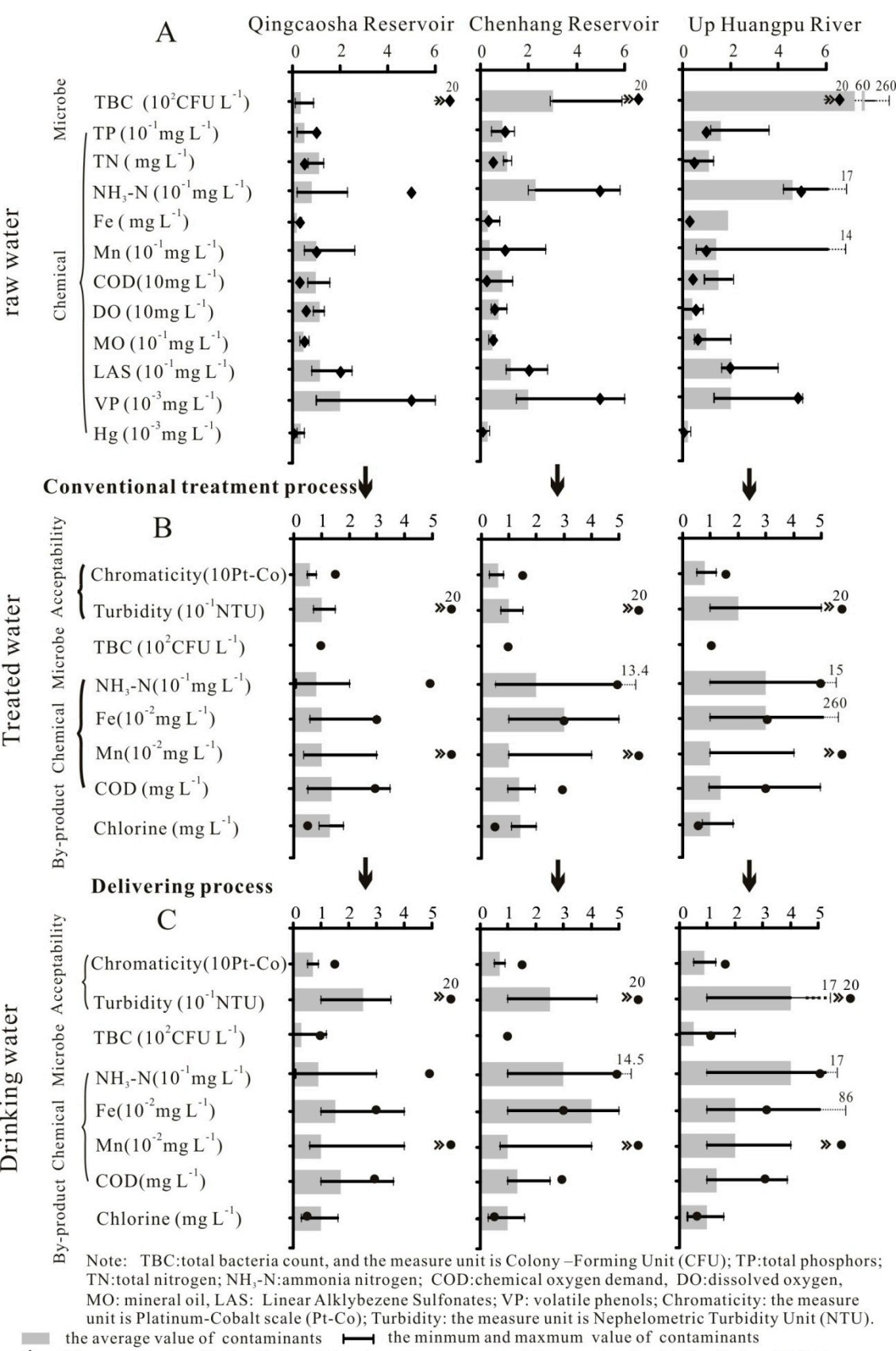

Note: TBC:total bacteria count, and the measure unit is Colony –Forming Unit (CFU); TP:total phosphors; TN:total nitrogen; NH₃-N:ammonia nitrogen; COD:chemical oxygen demand, DO:dissolved oxygen, MO: mineral oil, LAS: Linear Alklybezene Sulfonates; VP: volatile phenols; Chromaticity: the measure unit is Platinum-Cobalt scale (Pt-Co); Turbidity: the measure unit is Nephelometric Turbidity Unit (NTU).

▬ the average value of contaminants ⊢—⊣ the minmum and maxmum value of contaminants

◆ Chinese national guideline limits of II grade surface water ● Chinese national guideline limits of drinking water

**Figure 2.** The quality of water from the Qingcaosha Reservoir, the Chenhang Reservoir and the Huangpu River as it proceeds from raw water (**A**) to treated water (**B**) and at its final delivery point (**C**).

## 4. Results

### 4.1. Variations in the Quality of Raw Water, Treated Water and Drinking Water

The maximum values of the 10 priority contaminants of raw water sources of the Qingcaosha Reservoir, Chenhang Reservoir, and Huangpu River generally exceeded the national standard for surface water (Figure 2A). Although these contaminants decreased after treatment, the concentration of $NH_3$–N, Fe, COD, and chlorine exceeded the Chinese National Standard for drinking water (Figure 2A,B). Furthermore, after water has been delivered to households, some contaminants, such as TBC, $NH_3$–N, Fe, COD, and turbidity have increased in concentration (Figure 2B,C).

The raw water quality heavily influences the quality of household drinking water. The mean values of the 10 priority contaminants at the Qingcaosha Reservoir were generally lower than those of the Chenhang Reservoir, and the Huangpu River has the poorest overall quality water. Mean values of seven contaminants (TBC, TP, $NH_3$–N, Fe, Mn, COD, and LAS) in the Huangpu River were at least double those of the Qingcaosha and Chenhang Reservoirs and exceeded the national standard of grade II water (Figure 2A). In treated water and drinking water, the mean values of the eight priority contaminants of the Qingcaosha Reservoir were less than those of the Chenhang Reservoir and the Huangpu River (Figure 2B,C).

### 4.2. Variation of Eutrophication in the Changjiang Estuary and the Huangpu River

The TN, TP, and COD levels in the Huangpu River were generally at least twice those of the reservoirs in the Changjiang Estuary, exceeding the national standard of grade II water (Figure 3A–C). TN, TP, and COD increased before the mid-1990s and then decreased; however, TN, TP, and COD of the Changjiang Estuary have continually increased since the 1980s (Figure 3A–C). Of note, algal blooms occurred in all freshwater sources in the last decade. Algal density in the Qingcaosha Reservoir has increased to $12 \times 10^6$ L$^{-1}$ since it began operation in December 2010 (Figure 3D). Remarkably, the algal density of the Chenhang Reservoir reached $30 \times 10^6$ L$^{-1}$ in May 2003 (Figure 3E), and that of the Huangpu River increased to $6 \times 10^6$ L$^{-1}$ in the summer of 2005 (Figure 3F).

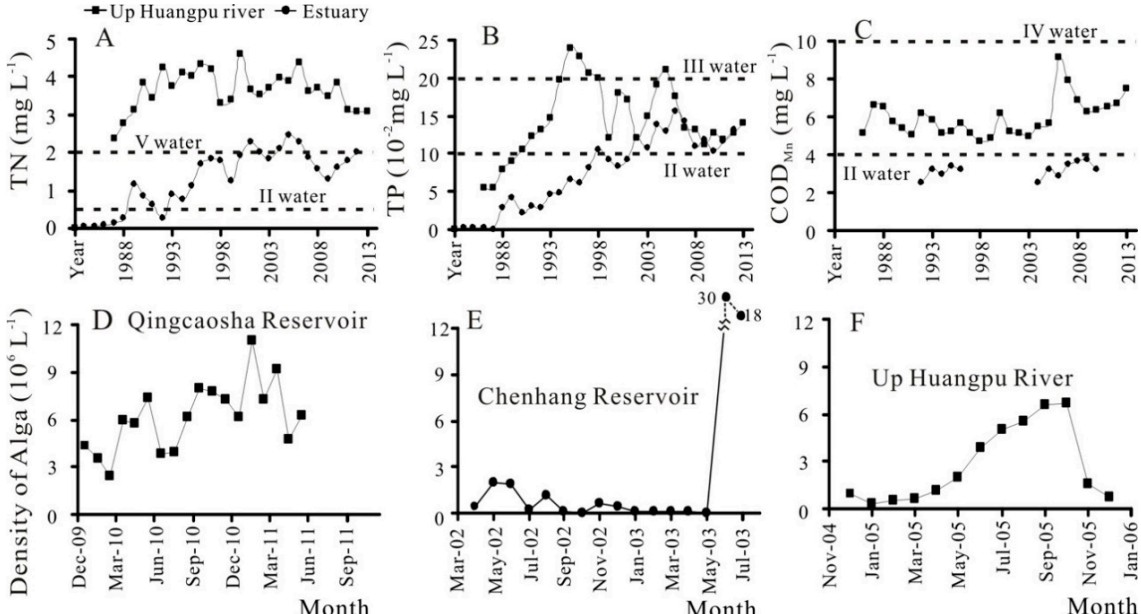

**Figure 3.** Eutrophication variations in the Changjiang Estuary and the Huangpu River. (**A**) Total nitrogen (TN) variations in 1984–2013; (**B**) TP variations in 1984–2013; and (**C**) COD$_{Mn}$ (as determined by potassium permanganate method) variations in 1984–2013. Also presented are the algal densities in the Qingcaosha Reservoir (**D**), and the Chenhang Reservoir (**E**), and the Huangpu River (**F**).

### 4.3. Salinity of Saltwater Intrusion Events in the Changjiang Estuary

Salinity in the three estuarine salinity survey stations that are shown in Figure 4 increased as discharge fell in every saltwater intrusion event in the low flow season. The salinity generally increased from 0 to 6 psu as discharge fell from 20,000 m³·s⁻¹ to 6700 m³·s⁻¹ as recorded at Datong station, the lowest hydrological station on the Changjiang River. Increasing salinity is closely related to lower discharge, especially when it was below 15,000 m³·s⁻¹, as demonstrated by the high values of explained variance in the regressions in Figure 4II.

The maximum salinity at Chongtou, Chenhang and Gaoqiao were 5.87 psu, 5.06 psu and 5.51 psu which corresponded to discharges of 10,000 m³·s⁻¹, 9600 m³·s⁻¹ and 6700 m³·s⁻¹, respectively (Figure 4). However, when discharge was in the range 15,000–20,000 m³·s⁻¹, the salinity at Gaoqiao was mainly below 0.45 psu, some occurrences of higher psu occurred at Chenhang, even up to 1 psu, while frequent occurrences of salinity above 0.45 and up to 2 psu occurred at Chongtou, even at discharges that were close to 20,000 m³·s⁻¹ (Figure 4). We discuss this pattern in Section 5.3, below.

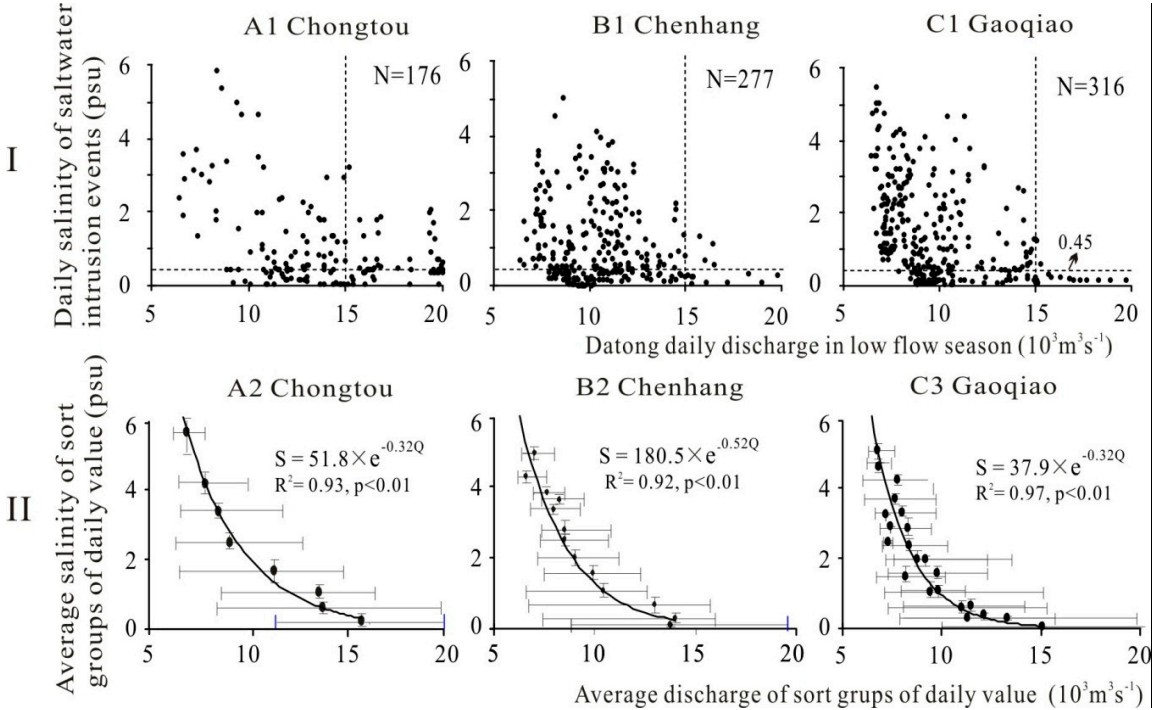

**Figure 4.** (**I**) Daily salinity levels at the salinity survey stations in relation to daily discharge (as recorded at Datong (with a seven-day lag for travel time to the estuary). Of note: while discharge >15,000 m³·s⁻¹, saltwater intrudes mainly from the North Branch, whereas discharge <15,000 m³·s⁻¹, saltwater comes from both the South and the North Branches. (**II**). The correlation equations between average salinity of sort groups and relevant average discharge of sort groups based on daily data, the salinity ranges of every sort group is 0.30 psu.

### 4.4. Removal Rates of Contaminants in Treated Water

The ability of water treatment plants to remove contaminants is demonstrated here while using the removal rate of two water plants, the Yuepu Water Plant, processing water from the Chenhang Reservoir on the Changjiang Estuary, and the Minghang No. 2 Water Plant, sourcing water from the Huangpu River at Songpu (Figures 1 and 5). The removal rates are shown in %, and negative values indicate increase, rather than removal. The outcomes can be divided into three levels (Figure 5). The removal of chromaticity, turbidity, TBC, TP, Fe, and Mn was highly efficient with rates of 50% or greater at both plants. There is low efficiency in the removal of COD, LAS, and VP, with removal rates ranging from 0% to 50%. The lowest efficiency of removal was for TN, chlorine, chloroform,

NH$_3$–N, and carbon tetrachloride, and had removal rates ranging from 0% to −100% (Figure 5). Note that chlorine is added to provide disinfection during the treatment process and also provides ongoing protection for the water while it is being distributed through the pipe network. However, if the level of Residual Chlorine exceeds 0.05 mg L$^{-1}$, it becomes a contaminant under the Chinese standards for drinking water quality [14]. This occurs when the chlorine demand of the water to be treated is excessively high because of the poor quality of the raw water.

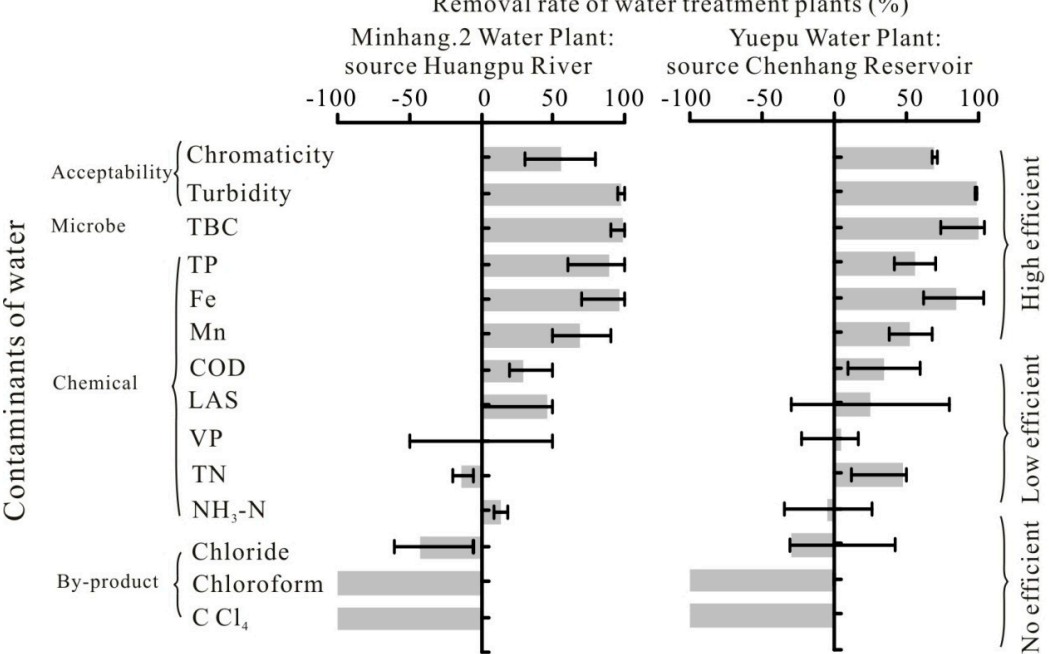

**Figure 5.** The removal rates 14 contaminants at two typical water treatment plants: the Minhang No. 2 water plant, and the Yuepu water plant. Negative values indicate increase, not removal.

*4.5. Changes to Chlorine, TBC, COD and Turbidity Levels in Delivery Pipes*

Changes to water quality are here shown in Figure 6 for water distributed by pipeline over a distance of 21 km from the Minhang No 2 Treatment Plant. Even though Chlorine decreases from 1.9 mg L$^{-1}$ to 1.1 mg L$^{-1}$ during delivery it still exceeds the maximum level of the Chinese national guidelines at the point of consumption (Figure 6A). TBC, COD and turbidity levels increased during delivery, with TBC increasing from 11 × 10$^3$ L$^{-1}$ to 45 × 10$^3$ L$^{-1}$, COD increased from 1.2 mg L$^{-1}$ to 3.5 mg L$^{-1}$, and turbidity increased from 0.09 NTU to 0.3 NTU (Figure 6B–D).

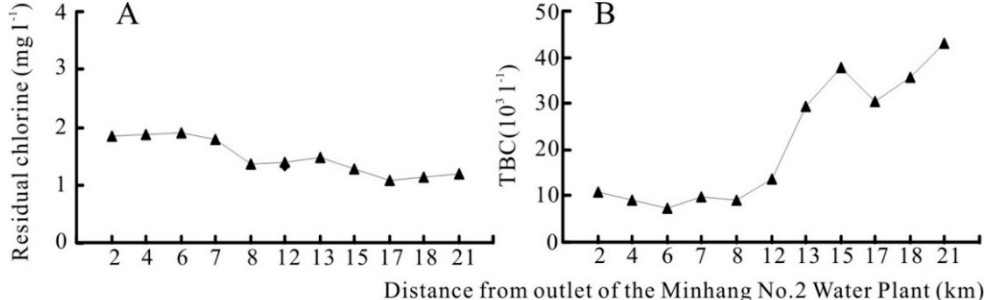

**Figure 6.** *Cont.*

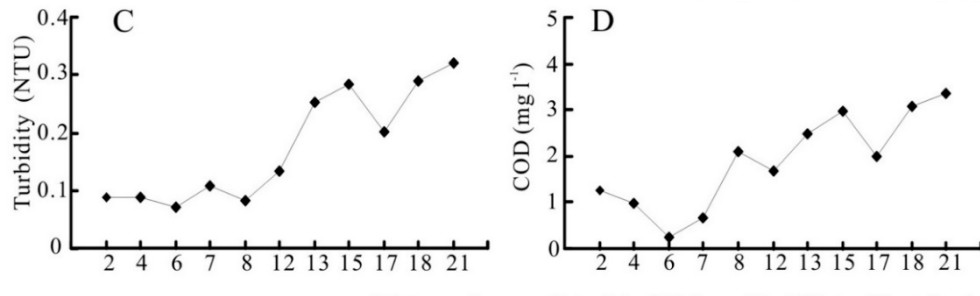

**Figure 6.** The deterioration of water quality during delivery via pipeline. (**A**) decreasing residual chlorine, (**B**) increasing bacteria, (**C**) increasing chemical oxygen demand, and (**D**) increasing turbidity.

## 5. Discussion

### 5.1. Water Quality: Pollutants in Raw Water

The maximum values of 12 major contaminants at all raw water sources generally exceeded the Chinese National Standard of grade II water, and therefore indicate that all of the freshwater sources for Shanghai do not meet the national standard (Figure 2A). The quality of the raw water has been greatly improved by conventional water treatment, in terms of meeting the national standards [14]) (Figure 2A,B). However, the maxima of TN, $NH_3$–N, Fe, COD, and Chlorine in treated water and drinking water still exceeded the national standards [14] (Figure 2B). This indicates that the conventional treatment technology of the current water plants is inadequate in Shanghai. Furthermore, the quality of the treated water is being degraded during delivery through the pipe network due to the poor quality of the distribution system. (Figure 2B,C).

Water quality at the Qingcaosha and Chenhang Reservoirs was better than that of the Huangpu sources (Figure 2A). The Changjiang Estuary, as a freshwater source, is far better than the Huangpu River. This is also confirmed by comparing the levels of eight contaminants in treated water and drinking water (Figure 2B,C). The values of many contaminants in the Huangpu River were three times higher than in the Qingcaosha Reservoir. This suggests that the Huangpu River is really no longer suitable as a source of raw water (Figure 2), and explains why the Shanghai government has been progressively shifting to sourcing its raw water from the Changjiang Estuary [46].

In Shanghai, groundwater was used as a raw water source in the past, but it has been phased out because it was causing ground subsidence with average subsidence rates of 40 mm per year in the central city [11,47,48]. We have not included groundwater as a source of drinking water in this study, as groundwater extraction has now virtually ceased and is being replaced by groundwater recharge to further help reduce subsidence.

### 5.2. Eutrophication Threats to Raw Water

Eutrophication is known as one of the key threats to water quality globally. Nitrogen and phosphorus are the two essential factors that drive eutrophication [42,43]. Increasing concentrations of TN, TP, and COD in the Changjiang Estuary and the Huangpu River indicate that the freshwater resources of Shanghai have remained at a high level of eutrophication since the 1980s when the economic boom began (Figure 3A–C) [49,50]. While TN and TP levels continue to be high in the Huangpu River raw water, they have actually declined since the mid-1990s, while they continue a steady overall increase in the Changjiang Estuary (Figure 3A,B). Given the shift towards sourcing water from the estuary, these trends are of some concern. Xu et al. (2013) pointed out that the increasing levels of dissolved inorganic nitrogen (DIN) can be attributed to increasing levels of untreated sewage being disposed of in the Changjiang. Their predictions of future DIN concentrations indicate that concentrations would range from 2.2–3.0 mg $L^{-1}$ over the period 2020–2050 [50]. This far exceeds

the 2.0 mg L$^{-1}$ maximum that is defined by the national standards and indicates the need for greatly improved sewage treatment practices in the Changjiang catchment.

As the levels of TN and TP increase, algal blooms have occurred in all Shanghai raw water sources (Figure 3D–F). Generally, blue - green alga (cyanobacteria) is dominant [51,52], especially in the lakes in the Changjiang delta, such as Taihu, which is the source of the Huangpu River [53]. Blue-green algae not only metabolically produce toxins such as microcystin and anatoxin-a, which can damage the liver, intestines, and nervous system if ingested [54], but also metabolically produce taste and odour components, such as geosmin and methylisoborneol, which affect drinking water [55]. Furthermore, algal blooms also produce many pathogenic bacteria and cause many water-related diseases [52]. However, little has been published regarding the algal bloom hazards of the Changjiang estuarine reservoirs. The Qingcaosha Reservoir has a water surface area of 70 km$^2$, a mean depth of only 6.2 m and a long-term storage time of 68 days. Such conditions readily initiate algal blooms in the high temperatures of the summer and autumn seasons, and the maximum algal density has reached up to $12 \times 10^6$ L$^{-1}$ (Figure 3D). Although the maximum microcystin concentration (0.55 µg/L) did not exceed the maximum that the WHO drinking water standard specified (1 µg/L), the extreme concentration of methylisoborneol (18 ng/L) has exceeded the threshold for odour (10 ng/L) and caused taste issues in drinking water [31]. Therefore, algal blooms have become a severe potential threat to the safety of Shanghai's estuarine raw water reservoirs.

### 5.3. Salinity Threat to Estuarine Raw Water

Salinity is a limiting factor in estuarine freshwater availability, as the water cannot be used in the treatment plants with salinity >0.45 psu [56–58]. The salinity variations of the three stations that are shown in Figure 4 clearly illustrate saltwater intrusions in relation to discharge. While discharge remains below 15,000 m$^3$·s$^{-1}$, the maximum salinity in downstream sequence of 5.87 psu at Chongtou >5.00 psu at Chenhang <5.51 psu at Gaoqiao (Figure 4) shows that the salinity at the three reservoirs resulted not only from the saltwater intrusion coming up the South Branch of the main estuary, but also from the North Branch around the upstream end of Chongming Island, as shown by the black arrows on the estuary in Figure 1B [58,59]. While discharge remained between 15,000–20,000 m$^3$·s$^{-1}$, the salinity gradually decreased from upstream Chongtou downstream to Chenhang and Gaoqiao (Figure 4). This indicates that the salinity of the three reservoirs mainly results from the saltwater intrusion coming around the North Branch [35,58,59]. The North Branch saltwater intrusion causes the Dongfengxisha Reservoir and the Chenhang Reservoir (15% of the total water supply capacity) to be unable to supply raw water to Shanghai for most of the dry season (about 126 days), when the daily discharge is less than 20,000 m$^3$·s$^{-1}$ [45].

The Qingcaosha Reservoir (IV in Figure 1) now supplies 57.6% of the total water supply to Shanghai and it draws on flow with the same salinity pattern, as shown in Figure 4, for the Gaoqiao salinity survey station with very high values, especially during the extreme low flow season. There is a probability of an increase in periods of extreme low discharge as about 2000 m$^3$·s$^{-1}$ will be potentially diverted for other cities along the mid-lower Changjiang in the next 50 years, water is now being diverted into the east branch of the south to north water transfer scheme, while the local sea level rise in the river estuary of 2 mm y$^{-1}$ will also increase the frequency and severity of saline intrusions [60,61]. Given this, at Qingcaosha by 2040 maximum salinity could reach 9 psu and the duration of saltwater intrusions could reach 124 days [45] (This is would mean that the capacity of the Qingcaosha Reservoir would be inadequate, as it can only store 68 days of raw water supply for Shanghai.

### 5.4. Inadequacy of the Present Water Treatment Process

The high removal rates demonstrate that the present water treatment technologies are adequate for removing inorganic and biological contaminants. However, the low removal rates indicate inadequate technology for removing organic and integrated contaminants, such as COD, NH$_3$–N, detergent (LAS), and volatile phenols (Figure 5). The concentrations of COD and NH$_3$–N in raw water sources

indicate that all Shanghai freshwater sources are polluted by organic materials and domestic sewage (Figure 2). The conventional water treatment processes many produce disinfection by-products, such as chlorine, chloroform, and carbon tetrachloride (Figure 5). The high concentrations of COD, $NH_3$–N and by-products in drinking water indicate that Shanghai is facing threats from organic and domestic contaminants. This is also a global issue, as freshwater becomes more polluted by anthropogenic organic contaminants and disinfection by-products [51,52].

The conventional water treatment in Shanghai generally consists of retention-based treatments (chemical coagulation, flocculation, and filtration) and degradation-based treatment (chlorination disinfection) (Figure 1A). The inefficiency of removal of COD is due to the present conventional retention-degradation technologies that are unable to eliminate the micro and lower molecular weight organics [24,62]. It is noted that many advanced retention-based technologies, such as activated carbon, membranes, and advanced UV irradiation and ozonation, have been applied in many developed countries [44,63,64] to remove micro organics and eliminate the disinfection by-products [15,65]. Currently, there are only two water treatment plants in Shanghai that have experimented with advanced water treatment technologies. Therefore, it is urgent that a long-term strategy for the improvement of Shanghai water treatment processes needs to be developed.

In addition to these problems with standards of treatment, it is also the case that Shanghai has a deficiency in treatment capacity that will become more severe as its population continues to grow. Li et al. (2017) constructed three scenarios of future water needs in Shanghai based on predictions of GDP and population growth to 2050 and showed that water treatment capacity will need to increase by between 35% and 83% beyond present levels [66].

### 5.5. Deterioration of Drinking water

The decrease in chlorine levels along water delivery pipes reduces its disinfection capability, while the increase in total bacterial count, COD, and turbidity along the pipes demonstrates the deterioration of drinking water quality during delivery (Figure 6A). As the disinfection capability declines, there is a consistent rise in total bacterial count, turbidity and COD (Figure 6B–D). The deterioration of drinking water quality would generally cause an increase in microbial and chemical contamination [67].

Bacteria counts increase along the pipe, not only because the existing nitrogen and phosphorus removal rates in the treatment plants are below 90%, but also in relation to the decrease in chlorine disinfection (Figures 5 and 6). Bacterial growth breeds worms, such as red worm (*limnodrilus hoffmeisteri*) in drinking water, which is one of the most severe problems [25] The bacterial increase also causes COD to increase along the pipe (Figure 6D), and further develops taste and odour components, such as geosmin and methylisoborneol, which give drinking water a bad smell [55]. Excessive chlorine is added to drinking water in order to prevent bacterial and odour deterioration, which explains the obvious smell of disinfection in household drinking water in Shanghai. Better treatment and higher quality raw water would reduce the need for such high chlorine levels.

The turbidity increase along the pipe is partly caused by increased bacteria, but it is mostly sourced from the corrosion and erosion of concrete and cast-iron pipes and tanks that accounts for 60% of all water supply pipes and tanks in Shanghai. Replacing the ageing concrete and cast-iron pipes and tanks by new steel or plastic composites could gradually improve the household drinking water quality.

## 6. Conclusions

Water quality for supply to urban areas is determined in terms of common standards of water constituents that ensure the water is safe for human consumption. The water supply process for most cities consists of three stages: raw water, treated water, and drinking water designed to eliminate pollutants. In this study, we have dealt with these three stages and the nature of contamination at each stage.

Many of the contaminants that have been reported here have concentrations in the Huangpu River raw water double that of water from the Changjiang Estuary. This has caused the Shanghai government

to reduce its historic dependence on the Huangpu River for raw water and shift its raw water intake to the Changjiang estuary to meet the clean water demands of the metropolitan population of 24 million. However, the Changjiang Estuary, like the Huangpu River, also has water quality problems with eutrophication, organic pollution, and algal blooms. Furthermore, saltwater, particularly that entering the estuary via the North Branch around Chongming Island, threatens the ability of the estuarine reservoirs to supply water that is suitable for treatment, while the Changjiang discharge is in the range 15,000–20,000 $m^3 \cdot s^{-1}$, and saltwater in the South Branch would exacerbate salinity while the discharge is below 15,000 $m^3 \cdot s^{-1}$, especially for the Qingcaosha Reservoir that supplies 56.7% of Shanghai's raw water.

Water treatment plants efficiently remove large amounts of inorganic and biological contaminants. However, more than 50% of organic and domestic contaminants, such as COD, $NH_3$–N, and some disinfection by-products, such as chlorine, are still retained in treated water and household drinking water. In addition, bacteria and increased turbidity reappear in the delivery pipes and tanks due to the decrease of residual chlorine and corrosion of cast-iron and concrete pipework and tanks. Therefore, Shanghai must efficiently remove organic contaminants by upgrading to advanced treatment technologies, such as activated carbon and ozonation, and prevent drinking water deterioration during delivery by renewing the old pipe and tank systems. This is a challenge for Shanghai, as it involves huge investment and complicated construction procedures in this high density city. The Shanghai government alone cannot deal with all of the problems, as increasing levels of untreated or poorly treated sewage are being put into the river as urbanization increasingly concentrates the population within the catchment.

**Author Contributions:** Conceptualization and methodology, M.L. and J.C.; investigation, M.L., J.C., B.F., Z.C., M.W. (Michael Webber), J.B. and M.W. (Mark Wang); writing—original draft preparation, M.L.; writing—review and editing, B.F., J.B., M.W. (Mark Wang) and J.C.; project administration, Z.C.; funding acquisition, M.W. (Michael Webber).

**Funding:** This research is funded by the Ministry of Science and Technology of the People's Republic of China grant number 2017YFC0506000 and 2016YFE0133700, and also be funded by the Australian Research Council, grant number P110103381.

**Acknowledgments:** The authors would like to thank Daowei Yin for assistance with checking manuscript, and also thank the anonymous reviewers for their valuable comments.

**Conflicts of Interest:** The authors declare no conflict of interest.

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
