# Peer review of "Freshwater Supply to Metropolitan Shanghai: Issues of Quality from Source to Consumers"

_water, doi:10.3390/w11102176_

Round 1
Reviewer 1 Report
Please see attached file for review comments.

Reviewer 2 Report
The manuscript is interesting, well-structured, but relatively limited in terms of proposals / conclusions.
Figures 2 and 5 should be clarified by indicating in the caption the meaning of the black strip and the gray band.
The main doubt regarding this work is its classification as Article. A classification as Communication, for example, seems more appropriate as the contribution of relevant scientific innovation is reduced.
Round 2
Reviewer 1 Report
The revised manuscript adequately addressed my review comments.